# Combined Buckwheat d-Fagomine and Fish Omega-3 PUFAs Stabilize the Populations of Gut *Prevotella* and *Bacteroides* While Reducing Weight Gain in Rats

**DOI:** 10.3390/nu11112606

**Published:** 2019-10-31

**Authors:** Mercè Hereu, Sara Ramos-Romero, Roser Marín-Valls, Susana Amézqueta, Bernat Miralles-Pérez, Marta Romeu, Lucía Méndez, Isabel Medina, Josep Lluís Torres

**Affiliations:** 1Institute for Advanced Chemistry of Catalonia (IQAC-CSIC), E-08034 Barcelona, Spain; merce.hereu@iqac.csic.es (M.H.); roser.marin@iqac.csic.es (R.M.-V.); joseplluis.torres@iqac.csic.es (J.L.T.); 2Department of Cell Biology, Physiology & Immunology, Faculty of Biology, University of Barcelona, E-08028 Barcelona, Spain; 3Departament d’Enginyeria Química i Química Analítica and Institut de Biomedicina (IBUB), Universitat de Barcelona, E-08028 Barcelona, Spain; samezqueta@ub.edu; 4Facultat de Medicina i Ciències de la Salut, Universitat Rovira i Virgili, E-43201 Reus, Spain; bernat.miralles@urv.cat (B.M.-P.); marta.romeu@urv.cat (M.R.); 5Instituto de Investigaciones Marinas (IIM-CSIC), E-36208 Vigo, Spain; luciamendez@iim.csic.es (L.M.); medina@iim.csic.es (I.M.)

**Keywords:** Weight, gut microbiota, *Prevotella*, Bacteroides, d-fagomine, iminosugar, iminocyclitol, ω-3 PUFAs

## Abstract

Some functional food components may help maintain homeostasis by promoting balanced gut microbiota. Here, we explore the possible complementary effects of d-fagomine and ω-3 polyunsaturated fatty acids (ω-3 PUFAs) eicosapentaenoic acid/docosahexaenoic acid (EPA/DHA 1:1) on putatively beneficial gut bacterial strains. Male Sprague–Dawley rats were supplemented with d-fagomine, ω-3 PUFAs, or both, for 23 weeks. Bacterial subgroups were evaluated in fecal DNA by quantitative real-time polymerase chain reaction (qRT-PCR) and short-chain fatty acids were determined by gas chromatography. We found that the populations of the genus *Prevotella* remained stable over time in animals supplemented with d-fagomine, independently of ω-3 PUFA supplementation. Animals in these groups gained less weight than controls and rats given only ω-3 PUFAs. d-Fagomine supplementation together with ω-3 PUFAs maintained the relative populations of *Bacteroides*. ω-3 PUFAs alone or combined with d-fagomine reduced the amount of acetic acid and total short-chain fatty acids in feces. The plasma levels of pro-inflammatory arachidonic acid derived metabolites, triglycerides and cholesterol were lower in both groups supplemented with ω-3 PUFAs. The d-fagomine and ω-3 PUFAs combination provided the functional benefits of each supplement. Notably, it helped stabilize populations of *Prevotella* in the rat intestinal tract while reducing weight gain and providing the anti-inflammatory and cardiovascular benefits of ω-3 PUFAs.

## 1. Introduction

There is some evidence that dietary components such as soluble fiber, polyphenols and polyunsaturated fatty acids (PUFAs) may have functional effects that protect against metabolic disorders and cardiovascular diseases [1,2,3,4]. It is also becoming increasingly evident that some of these effects are mediated by changes in gut microbiota [5]. Diet has been shown to influence microbiota in both animal models and humans [6,7,8]. Although the gut microbiota is relatively stable in healthy adults [9], short-term disturbances can rapidly change its composition [10,11] with unknown effects on the host organism.

d-Fagomine (1,2-dideoxynojirimycin) is a six-ring iminocyclitol: a carbohydrate analog with a nitrogen atom in place of the endocyclic oxygen. d-Fagomine was first isolated from seeds of buckwheat (*Fagopyrum esculentum*) and it is also present in other plant parts such as mulberry (*Morus alba*) leaves and gogi (*Lycium chinense*) roots [12,13]. The functional effects of d-fagomine include a reduction of post-prandial blood glucose concentration, achieved through the inhibition of intestinal disaccharidases [14]; and reductions in high-fat-diet-induced weight gain, low-grade inflammation and impaired glucose tolerance, probably all achieved by counteracting adverse changes in gut microbiota [15,16]. Moreover, d-fagomine promotes the diversity of gut microbiota by increasing populations of Bacteroidetes in healthy rats while mitigating the age-related reduction in the populations of putatively beneficial *Lactobacillus* and *Bifidobacterium* bacteria [17].

Eicosapentaenoic acid (EPA, 20:5, *n*-3) and docosahexaenoic acid (DHA, 22:6, *n*-3) are the major ω-3 PUFAs of marine origin. EPA and DHA are essential dietary components that could help to reduce risk factors (plasma cholesterol and triglycerides, oxidative stress, and blood pressure) for cardiovascular diseases [4,18], and other pathologies that involve inflammation [19]. ω-3 PUFAs are believed to exert their anti-inflammatory effects by competing with arachidonic acid (ARA) metabolism and fostering the synthesis of anti-inflammatory mediators such as resolvins [19]. However, the effects of ω-3 PUFAs on gut microbiota are poorly documented [20]. It has been reported that EPA and DHA significantly increase the populations of Firmicutes (*Lactobacillus* genus) and Bifidobacteria in mice fed a high-fat diet [21,22]. In contrast, we found that the mixture EPA/DHA 1:1 reduced the population of Lactobacilliales in Wistar–Kyoto (WKY) rats fed a standard diet [23]. As buckwheat d-fagomine and ω-3 PUFAs have different effects on *Lactobacillus* spp. and *Bifidobacterium* spp., we decided to test whether their combination had any additive or complementary effects. We previously reported that a combination of EPA/DHA 1:1 and proanthocyanidins can be instrumental in promoting balanced gut microbiota [23]. In this paper, we now also focus on the populations of *Prevotella*, because this genus has been directly associated with improved glucose metabolism in humans [24], which is ultimately one of our main interests. We measured different variables in test animals (fat accumulation, weight gain, lipid profile and inflammation markers) that are pertinent to the known effects of d-fagomine and ω-3 PUFAs. Other studies have examined the effects of supplements on animal models subjected to more or less severe dietary challenges such as high-fat or high-sugar loads. As functional food components are primarily supposed to maintain the normal functions of the body, here we chose to test the effects on normal rats fed a standard diet. Therefore, the goal of this study was to explore the possible complementary functional effects of d-fagomine and EPA/DHA 1:1 on healthy rats.

## 2. Materials and Methods 

### 2.1. Animals and Diets

Male Sprague–Dawley (SD) rats (*n* = 36) from Envigo (Indianapolis, IN, USA), aged 10–11 weeks were housed (*n* = 3 per cage) under controlled conditions of humidity (60%), and temperature (22 ± 2 °C) with a 12 h light–12 h dark cycle. Prior to the nutritional intervention, the animals were fed a standard diet (2014 Teklad Global 14% Protein Diet from Envigo) *ad libitum* with free access to water (Ribes, Barcelona, Spain) for two weeks. Then they were divided into 4 groups (*n* = 9 per group): the control (CTL) group fed only the standard diet; a group supplemented with d-fagomine (FG); a group supplemented with ω-3 PUFAs (EPA/DHA 1:1) (ω-3); and a group supplemented with both d-fagomine and ω-3 PUFAs (FG + ω-3). d-Fagomine (>98%) manufactured by Bioglane SLNE (Barcelona, Spain) was generously provided by Taihua Shouyue (HK) International Co. Ltd. (Hong Kong, China). It was included in the feed at a proportion of 0.96 g/kg feed as defined in previous studies [14,16]. The mean daily dose of d-fagomine was 4.6 mg per 100 g body weight, calculated from a mean feed consumption of 4.8 g feed per day per 100 g body weight. The mixture EPA/DHA 1:1 was obtained by mixing the appropriate quantities of the commercial fish oils AFAMPES 121 EPA (AFAMSA, Vigo, Spain) and EnerZona Omega 3 RX (EnerZona, Milan, Italy). These ω-3 PUFAs were administered by oral gavage using a gastric probe once a week at a dose of 0.8 mL oil per kg body weight. The dose and EPA/DHA proportions used were those reported previously [25]. To compensate for the stress of probing and the excess calories from the fish oil in groups ω-3 and FG + ω-3, the animals in groups CTL and FG were administered soy bean oil at the same dose and at the same time. 

All the procedures carried out strictly adhered to European Union Directive 2010/63/EU for the care and management of laboratory animals, and were licensed by the regional Catalan authorities (reference no. DAAM7921), as approved by the Spanish CSIC Subcommittee of Bioethical Issues.

### 2.2. Data and Sample Collection 

Feed consumption was monitored daily and body weight was measured weekly throughout the experiment. Based on feed intake, the mean daily dose of d-fagomine was 4.5 mg/100 g body weight. Energy intake was calculated as estimates of metabolizable energy based on the Atwater factors: 4 kcal/g protein, 9 kcal/g fat, and 4 kcal/g available carbohydrate. 

After week 21 of the experiment, fecal samples were collected by abdominal massage, immediately frozen and stored at −80 °C until analysis. To reduce circadian rhythm interference, samples were taken 2–3 h into the light period. Blood samples were collected from the saphenous vein after overnight fasting. Plasma was separated by centrifugation and stored at −80 °C until analysis. 

At the end of the experiment (week 23), rats were fasted overnight and anaesthetized intraperitoneally with ketamine and xylazine (80 and 10 mg/kg body weight, respectively). Blood was collected by cardiac puncture, then plasma was immediately obtained by centrifugation and stored at −80 °C until analysis. Perigonadal fat was collected, weighed and immediately frozen in liquid N_2_. All the samples were stored at −80 °C until analysis.

### 2.3. Glycemic Status

An oral glucose tolerance test (OGTT) was performed at week 18 on fasted animals. A solution of glucose (1 g/kg body weight) was administered by oral gavage before the test, and blood glucose concentration was measured 15, 30, 45, 60, 90 and 120 min after the glucose intake. Blood glucose concentration was measured by the enzyme electrode method, using an Ascensia ELITE XL blood glucose meter (Bayer Consumer Care, Basel, Switzerland).

Fasting blood glucose and plasma insulin levels were also measured after week 21, in fasted animals. Plasma insulin levels were measured using the rat/mouse insulin enzyme-linked immunosorbent assay (ELISA) kit from Millipore Corporation (Billerica, MA, USA). 

### 2.4. Plasma Lipid Profile

Plasma triglycerides, total cholesterol, high-density lipoprotein (HDL)-cholesterol and low-density lipoprotein (LDL)-cholesterol were measured using a spectrophotometric method and the corresponding kits from Spinreact (Girona, Spain) as described elsewhere [26,27]. 

### 2.5. Plasma Lipid Mediators of Inflammation

Several hydroxyeicosatetraenoic acids (HETEs), lipid mediators from the metabolism of ARA, were determined in plasma by liquid chromatography coupled with tandem mass spectrometry (LC-MS/MS) using a method modified from Dasilva et al. [28]. Erythrocyte-free plasma samples (90 µL) were thawed, diluted in the presence of butylated hydroxytoluene (BHT), and spiked with the internal standard (12HETE-d8), Cayman Chemicals, Ann Arbor, MI, USA). Then, the samples were centrifuged (800 g, 10 min), and the lipids in the supernatants were purified by solid-phase extraction (SPE). The LC-MS/MS analyzer consisted of a Dionex UltiMate 3000 Series chromatograph (Thermo Fisher, Rockford, IL, USA) coupled to a dual-pressure linear ion-trap mass spectrometer LTQ Velos Pro (Thermo Fisher, Rockford, IL, USA) operated in negative electrospray ionization (ESI) mode. A C18-Symmetry 150 × 2.1 mm inner diameter, 3.5 µm column (Waters, Milford, MA, USA) with a C18 4 × 2 mm guard cartridge (Phenomenex, Torrance, CA, USA) were used in the separation step. Samples (10 µL) were eluted with a binary system consisting of 0.02% aqueous formic acid [A] and 0.02% formic acid in methanol [B] under gradient conditions of: 0 min, 60% B; 2 min, 60% B; 12 min, 80% B; 13 min, 80% B; 23 min, 100% B; 25 min, 100% B; and 30 min, 60% B, at a flow rate of 0.2 mL/min

### 2.6. Fecal Microbial Populations

The relative populations of selected bacterial phyla, orders and genera (Bacteroidetes, Firmicutes, Lactobacilliales, Bifidobacteriales, *Prevotella*, and *Bacteroides*) were estimated in fecal DNA by quantitative real-time polymerase chain reaction (qRT-PCR). DNA was extracted from feces using a QIAamp^®^ DNA Stool Mini Kit from QIAGEN (Hilden, Germany) and its concentration was quantified using a Nanodrop 8000 Spectrophotometer (ThermoScientific, Waltham, MA, USA). qRT-PCR experiments were carried out in triplicate using a LightCycler^®^ 480 II (Roche, Basel, Switzerland). Each qRT-PCR well contained DNA (2 µL of a 20 ng/µL solution) and a master mix (18 µL) made of 2X SYBR (10 µL), the corresponding forward and reverse primer (1 µL each), and water (6 µL). All the reactions were paralleled by analysis of both a non-template control (Milli Q water) and a positive control (Table 1) from DSMZ (Braunschweig, Germany). The qRT-PCR cycling conditions were: 10 s at 95 °C, then 45 cycles of 5 s at 95 °C, 30 s at the primer-specific annealing temperature (Table 1), and 30 s at 72 °C (extension). The specificity of the qRT-PCR reactions was assessed by melting curve analysis which consisted of heating to 95 °C and maintaining this temperature for 2 s, then cooling to 65 °C and maintaining this temperature for 15 s, and running a temperature gradient from 65 °C to 95 °C at a rate of 0.11 °C/s, with five fluorescence recordings per °C. The relative DNA abundances for each bacterial subgroup were calculated from the second derivative maximum of their respective amplification curves (Cp, calculated in triplicate) by considering Cp values to be proportional to the dual logarithm of the inverse of the specific DNA concentration, following the equation: [DNAa]/[DNAb] = 2Cpb−Cpa [29]. Amounts of total bacteria were normalized as 16S rRNA gene copies per mg of wet feces (copies/mg). 

### 2.7. Fecal Short-Chain Fatty Acids

Short-chain fatty acids (SCFAs) were analyzed in fecal samples after 21 weeks of supplementation by gas chromatography using a previously described method [35] with some modifications. Briefly, the freeze-dried feces were weighed (~50 mg dry matter) and a solution (1.5 mL) containing the internal standard 2-ethylbutyric acid (6.67 mg/L) and oxalic acid (2.97 g/L) in acetonitrile/water 3:7 was added. Then, SCFAs were extracted for 10 min using a rotating mixer. The suspension was centrifuged (5 min, 12,880× *g*) in a 5810R centrifuge (Eppendorf, Hamburg, Germany) and the supernatant filtered through a 0.45 µm nylon filter. Then an aliquot of the supernatant (0.7 mL) was diluted with acetonitrile/water 3:7 to a final volume of 1 mL. SCFAs were analyzed using a Trace2000 gas chromatograph coupled to a flame ionization detector (ThermoFinnigan, Waltham, MA, USA) equipped with an Innowax 30 m × 530 µm × 1 µm capillary column (Agilent, Santa Clara, CA, USA). Chrom-Card software was used for data processing. Helium was used as carrier gas with a linear velocity of 5mL/min. GC oven temperature was programmed as follows: 80 °C (hold 1min) to 120 °C at 15 °C/min (hold 4 min) to 130 °C at 5 °C/min (hold 4 min) to 235 °C at 8 °C/min (hold 4 min). Flame ionization detector (FID) detection was performed at a base temperature of 240 °C. Calibration curves were prepared using seven matrix-matched standards covering the working concentration range. The precision relative standard deviation (RSD < 15%) and recovery (>70%) of the method were adequate and inter- and intra-day reproducible. 

### 2.8. Statistical Analysis

All data manipulation and statistical analysis were performed using GraphPad Prism 5 (GraphPad Software, San Diego, CA, USA). The results are expressed as means with their standard errors (SEM). The normal distribution and heterogeneity of the data were evaluated, and statistical significance was determined by one-way analysis of variance (ANOVA) with each group as variable and the Tukey multiple-comparisons test, two-way ANOVA for repeated measures of body weight and glycemic response or Student’s *t*-test to compare the populations of gut microbiota of the CTL group at week 21 vs. week 0. Differences were considered significant when *p* < 0.05. 

## 3. Results

### 3.1. Feed and Energy Intake and Body Weight

Feed and energy intake were similar for all the groups throughout the experiment (Table 2). In contrast, the animals in the FG and FG + ω-3 groups gained less weight than those in the CTL and ω-3 groups (Figure 1a). At the end of the study (23 weeks) the animals supplemented with d-fagomine had significantly (*p* < 0.05) lower body weight than the controls or those supplemented with only EPA/DHA 1:1 (Table 2). Animals supplemented only with d-fagomine showed a tendency (*p* = 0.06 FG vs. CTL) to store lower perigonadal fat than those not supplemented (Figure 1b). 

### 3.2. Glycemic Status and Plasma Lipid Profile

The areas under the curve from the OGTT between CTL and FG+ ω-3 groups were statistically different at week 18 (Table 2). Fasting blood glucose and plasma insulin concentrations were measured after week 21 of the intervention. The levels of fasting glucose were statistically similar in all the groups and below 80 mg/dL (Table 2). The animals supplemented with d-fagomine showed lower fasting blood insulin concentrations than rats supplemented only with ω-3 PUFAs (Table 2).

The levels of total triglycerides, cholesterol and both HDL- and LDL-cholesterol were measured in plasma after 23 weeks of the intervention (Table 2). The concentrations of plasma triglycerides, cholesterol and the HDL/LDL ratio were significantly (*p* < 0.05) lower in the three supplemented groups than the control values. 

### 3.3. Plasma Lipid Mediators of Inflammation

The levels of ARA-derived pro-inflammatory eicosanoids were measured by LC-MS/MS in plasma samples collected at the end of the study (Figure 2). The plasma concentrations of 11HETE and 20HETE (Figure 2b,e) were significantly (*p* < 0.05) reduced in the two groups supplemented with ω-3 PUFAs with respect to the control group. The concentration of 12HETE (Figure 2c) was significantly (*p* < 0.05) lower in the three supplemented groups than the control values and the concentration of 5HETE and 15HETE (Figure 2a,d) were significantly (*p* < 0.05) reduced in the FG+ω-3 group compared to the FG group.

### 3.4. Bacterial Subgroups of Gut Microbiota

In the CTL group, the proportion of Bacteroidetes significantly (*p* < 0.05) decreased (83 ± 10 at week 0, 39 ± 9 at week 21) and the proportion of Firmicutes significantly (*p* < 0.01) increased (9 ± 2 at week 0, 26 ± 3 at week 21) with time; the percentages of Bacteroidetes and Firmicutes were similar in all the groups at the end of the study (week 21).

Significant differences in the relative populations of the genera *Prevotella* and *Bacteroides* were detected when comparing the control group at different times and between groups at the end of the study (Figure 3d,e). The populations of both genera significantly decreased in the control group after 21 weeks (*Prevotella*: 32 ± 8 at week 0, 6 ± 2 at week 21; *Bacteroides*: 65 ± 11 at week 0, 13 ± 2 at week 21). The proportion of *Prevotella* was significantly (*p* < 0.05) higher in animals supplemented with d-fagomine (FG and FG + ω-3 groups) than in the other groups (Figure 3d). The percentage of *Bacteroides* was higher (*p* < 0.05) in animals supplemented with the combination of d-fagomine and ω-3 PUFAs; while neither single supplementation significantly modified the proportions of this genus on its own (Figure 3e).

The percentage of Lactobacillales was significantly (*p* < 0.05) higher in the ω-3 group than in the FG group at the end of the intervention (Figure 3g), while the population of Bifidobacteriales was significantly (*p* < 0.05) higher in the FG group than in the ω-3 group (Figure 3h).

### 3.5. Short-Chain Fatty Acids

ω-3 PUFA supplementation (ω-3 and FG + ω-3 groups) reduced the fecal acetate content and the total short-chain fatty acids with respect to the control groups (*p* < 0.001). The concentration of isobutyric acid was significantly (*p* < 0.001) lower in the three supplemented groups than the control values (Table 3).

## 4. Discussion

The present study focuses on the effect of the combination of d-fagomine and ω-3 PUFAs (EPA/DHA 1:1) on gut microbiota of SD rats fed a standard diet. Our goal was to assess the capacity of these supplements to maintain a healthy status over time. Measuring and discussing biologically significant effects of food components in normal rats (or humans) is a particularly difficult task because the metabolic changes experienced by adequately fed animals are small. In the present study, the animals in all the groups were normoweight with normal growth curves (Figure 1a) and presented normal values of fasting blood glucose (Table 2) throughout the whole experimental intervention (≈5 months). Some statistically significant changes were recorded that may offer clues as to the putative protective effects of the supplementations and their combination. A weekly single dose of EPA/DHA (1:1) did not modify weight gain compared to the CTL group (SD rats) (Figure 1a), in agreement with our previous observations [25] in spontaneously hypertensive obese (SHROB) rats, which are a cross between an SD male and a WKY female rat [36]. Another study by our group showed that intensive daily supplementation with EPA/DHA (1:1) slightly increased both weight gain and perigonadal fat in female WKY rats compared to the group supplemented with the same dose of soybean oil [23]. These results highlight the differences between rat strains and doses in terms of response to putatively obesogenic components [37]. d-Fagomine consistently reduced weight gain in the present study, as in previous reports [15,16]. We have now shown here that this iminosugar reduced body weight gain by 15% over the 5 months of the intervention, when administered either alone or together with ω-3 PUFAs. These results are in line with previous studies which proved that d-fagomine was capable of reducing body weight gain in both SD and WKY rats fed energy-dense diets [15,16,38]. Therefore, d-fagomine appears to be effective at reducing body weight gain in both SD and WKY rats fed either a standard or an obesogenic diet. The supplemented animals also showed reduced levels of plasma triglycerides and total cholesterol at the end of the intervention (Table 2). This reduction of plasma triglycerides as a result of d-fagomine supplementation agrees with our previous observations in rats fed an obesogenic diet [38]. The reduction in the plasma concentration of triglycerides and cholesterol resulting from ω-3 PUFA supplementation also agrees with previous studies where this treatment reduced the levels of total plasmatic fatty acids in healthy rats [27]. The effects of ω-3 PUFAs on the lipid profile seem to be related to the upregulation of the expression of genes encoding proteins involved in fatty acid oxidation and downregulation of genes encoding proteins necessary for lipid synthesis [39].

Inflammatory status is another variable that may be influenced by dietary habits and ageing. The anti-inflammatory effect of EPA and DHA in both humans and animal models of disease is well documented [19]. We show here that EPA/DHA 1:1 reduced the levels of several ARA-derived pro-inflammatory lipid mediators: 11HETE, 12HETE and 20HETE (Figure 2). This reduction may be explained by the displacement of the pro-inflammatory ω-6 pathway towards the ω-3 pathway, as both metabolic pathways share several oxygenases (cyclooxygenase and lipoxygenases) [19]. d-Fagomine only reduced the levels of pro-inflammatory mediator 12HETE. We have suggested that d-fagomine might be effective at the very early stages in the development of low-grade inflammation by a eubiotic effect of on gut microbiota [16] while ω-3 PUFAs would have an anti-inflammatory effect after the pro-inflammatory pathways have been activated [19]. The results presented here are suggesting that the anti-inflammatory activity of the combination between d-fagomine and EPA/DHA 1:1 can be ascribed mainly to the ω-3 PUFAs as d-fagomine only reduced the levels of 12HETE. It should be noted that the animals given only the iminosugar were administered a dose of soybean oil equivalent to the dose of PUFAs in the ω-3 groups. The amount of linoleic acid (ω-6, 51% in soybean oil) given to the animals in the FG group might have counteracted the putative anti-inflammatory activity of d-fagomine. This would explain the lack of activity in the FG group (mediators 5HETE, 11HETE, 15HETE and 20HETE) and the significant differences in the levels of 5HETE and 15HETE between the FG + ω-3 and the FG group. The combination of supplements in the absence of supplemented soybean oil resulted in a significant reduction in the levels of all the mediators tested (Figure 2).

As it is becoming increasingly evident that fat accumulation, low-grade inflammation and gut microbiota are all interconnected [35,40,41,42], we examined the changes in relevant bacterial groups experienced by our experimental animals. The population of Bacteroidetes, the main gut microbiota phylum, was seen to reduce significantly over time in the control animals and neither d-fagomine nor ω-3 PUFAs had any significant influence on it. As the supplements did not show any statistically significant effect at the phylum level, we examined the genera *Prevotella* and *Bacteroides*, which are major subgroups of Bacteroidetes. In humans, diets that are high in complex carbohydrates and dietary fiber have been associated with dominance of the genus *Prevotella*; whereas high fat/protein diets have been connected with higher levels of the genus *Bacteroides* [8,11]. Human subjects with a high *Prevotella/Bacteroides* ratio appear to lose more body fat when on diets that are high in fiber than subjects with a low *Prevotella/Bacteroides* ratio [43]. In agreement with this, the consumption of barley kernel-based bread resulted in both improved glucose metabolism and increased populations of *Prevotella*, particularly *P. copri* [24]. Those same authors also offered evidence of a cause and effect relationship between *Prevotella* and glucose metabolism efficiency in the host, as germ-free mice transplanted with microbiota from responders had improved glucose tolerance and showed increased populations of *Prevotella*, compared to mice given microbiota from non-responders [24]. Here, we report that the populations of *Prevotella* dropped significantly in SD rats over the 21-week period of the intervention in both the CTL and ω-3 groups. Meanwhile, theses populations remained stable in animals supplemented with d-fagomine, independently of ω-3 PUFA supplementation (Figure 3d). Those animals (the FG and FG + ω-3 groups) had the lowest body weight gain. As the proportion of *Prevotella* in gut microbiota is directly related to the intake of dietary fiber and to improved glucose tolerance [24], we suggest that d-fagomine exerts a fiber-like action which affects microbiota-related fat accumulation and weight gain. Both ω-3 PUFAs and d-fagomine appear to induce a slight increase of *Bacteroides* which was statistically significant in the case of the double supplementation (the FG + ω-3 group) (Figure 3e). The group supplemented only with d-fagomine presented a *Prevotella* to *Bacteroides* ratio that was significantly higher than those of the other groups (Figure 3f). Meanwhile, in the FG + ω-3 group, the *Prevotella/Bacteroides* ratio was not statistically different from that in the CTL group; this is probably because of the additive effect of d-fagomine and ω-3 PUFAs on the populations of *Bacteroides* (Figure 3e). The physiological significance of the observation that d-fagomine and ω-3 PUFAs may induce an increase in the populations of *Bacteroides* is something to be examined in future studies.

A reduction in the populations of some putatively beneficial bacteria such as Lactobacilli and Bifidobacteria is a risk factor for the development of many intestinal conditions, including diarrhea, obesity, irritable bowel syndrome and inflammatory bowel disease [44]. Numbers of bacterial species of the genera *Bifidobacterium* and *Lactobacillus* are negatively correlated with adiposity, microbe-derived inflammation and obesity [42,45]. Our results show that d-fagomine tended to promote the growth of Bifidobacteria, while ω-3 PUFAs tended to increase the populations of Lactobacilli (Figure 3g,h). These differences were statistically significant when the two individually supplemented groups were compared. The results for individual supplementation are in agreement with previous reports by us and others. For instance, ω-3 PUFAs increased the populations of Lactobacilliales and Bifidobacteriales while reducing ω-6 PUFA-induced inflammation in mice [46]; and d-fagomine partially counteracted the loss of these two groups in WKY rats over time [17]. We show here that combined supplementation with d-fagomine and EPA/DHA may contribute to host homeostasis by maintaining the relative populations of putatively beneficial Bifidobacteriales and Lactobacilliales at levels similar to those of the CTL group.

We also observed differences in fecal SCFAs, which are products of bacterial fermentation. The two groups supplemented with ω-3 PUFAs presented significantly lower concentrations of acetate (*p* < 0.001) (Table 3). It has been reported that acetate may counteract obesity-induced low-grade inflammation by upregulating anti-inflammatory regulatory T cells and by reducing the production of cytokines and chemokines [47,48,49]. As fecal acetate content was lower in EPA/DHA-supplemented animals, our results seem to imply that this variable does not contribute to the possible effect of ω-3 PUFAs on host homeostasis. A more careful examination of the literature may lead to the opposite conclusion, as it has been shown that low levels of acetate in feces are inversely correlated with intestinal absorption [50]; therefore, they may be an indication of higher bioavailability. In fact, high fecal acetate has been associated with gut dysbiosis, obesity and hypertension [51]. Similarly, lower levels of excreted SCFAs together with higher Bacteroidetes: Firmicutes ratios have consistently been associated with the lean healthy phenotype, compared to metabolically altered phenotypes [52]. Thus, the anti-inflammatory action of ω-3 PUFAs may in part be mediated by an increase in acetate absorption in the intestinal tract. As we did not record any significant intergroup differences in the populations of Firmicutes, which are supposed to include the main SCFA-producing gut microorganisms, we suggest that minor species may be affected by ω-3 PUFAs, independently of the action of d-fagomine. This is another point worth exploring in future studies. The main limitation of this study lies in the fact that differences between groups of healthy rats given dietary supplements are necessarily small and they hardly reach statistical significance. Nevertheless, the comparative analysis of the present results with other studies from our group using different rat strains, sex and diets is suggesting a consistent effect of d-fagomine on weight gain and glycaemia. The evaluation of gut microbiota and their fermentation products SCFAs presented here has also its limitations. Further studies should address more comprehensively the possible variations in bacterial subgroups, species and strains. The systemic concentration of SCFAs would provide a more accurate approximation to their effect that their levels in feces. The translation of the present results to humans is not straightforward as d-fagomine is not yet an approved dietary ingredient or new food. Human testing of its effects, whether alone or combined with ω-3 PUFAs, should use food sources such as buckwheat or mulberry, which contain amounts of the active compound that are too low [13].

## 5. Conclusions

This paper presents the first evidence that bacteria of the genus *Prevotella*, which are associated with functional effects on glucose metabolism, may mediate the microbiota-related effects of d-fagomine on host homeostasis. The combination between d-fagomine and ω-3 PUFAs, stabilizes the populations of putatively beneficial gut bacteria, and reduces weight gain and pro-inflammatory lipid mediators.

## Figures and Tables

**Figure 1 nutrients-11-02606-f001:**
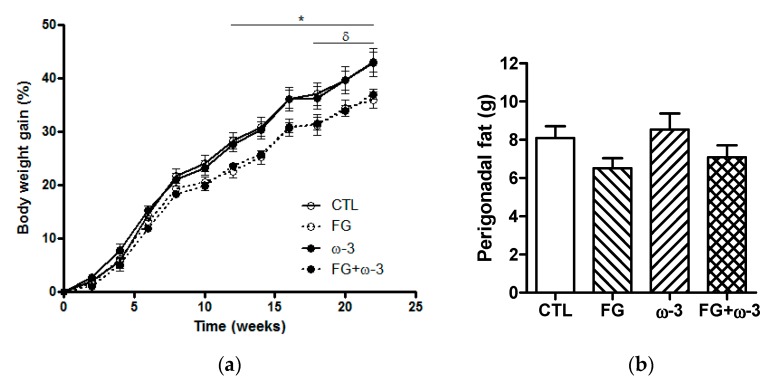
Body weight gain (**a**) and perigonadal fat (**b**) of the different groups (CTL, FG, ω-3 and FG + ω-3) of Sprague–Dawley rats fed a standard diet for 23 weeks. The data represent means with their standard errors. Comparisons were performed using two-way ANOVA for repeated measures (**a**), or one-way ANOVA followed by Tukey’s post-hoc test (**b**). **p* < 0.05 FG vs. CTL group; ^δ^*p* < 0.05 FG + ω-3 vs. CTL group.

**Figure 2 nutrients-11-02606-f002:**
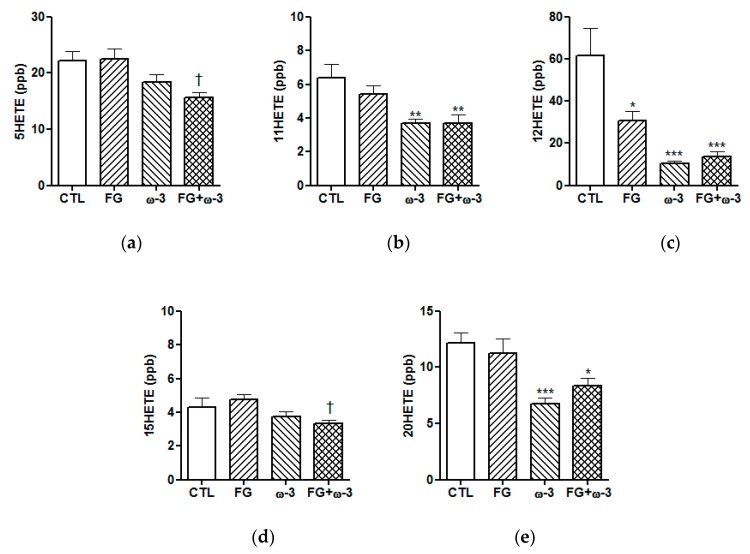
Lipid mediators from arachidonic acid (ARA): 5HETE (**a**), 11HETE (**b**), 12HETE (**c**), 15HETE (**d**) and 20HETE (**e**) in plasma, at the end of the study of the different groups (CTL, FG, ω-3 and FG + ω-3) of Sprague-Dawley rats fed a standard diet for 23 weeks. Data are presented as means with their standard error. Comparisons were conducted using one-way ANOVA and Tukey’s multiple comparisons test. **p* < 0.05, ***p* < 0.01, ****p* < 0.001 vs. CTL; ^†^*p* < 0.05 vs. FG.

**Figure 3 nutrients-11-02606-f003:**
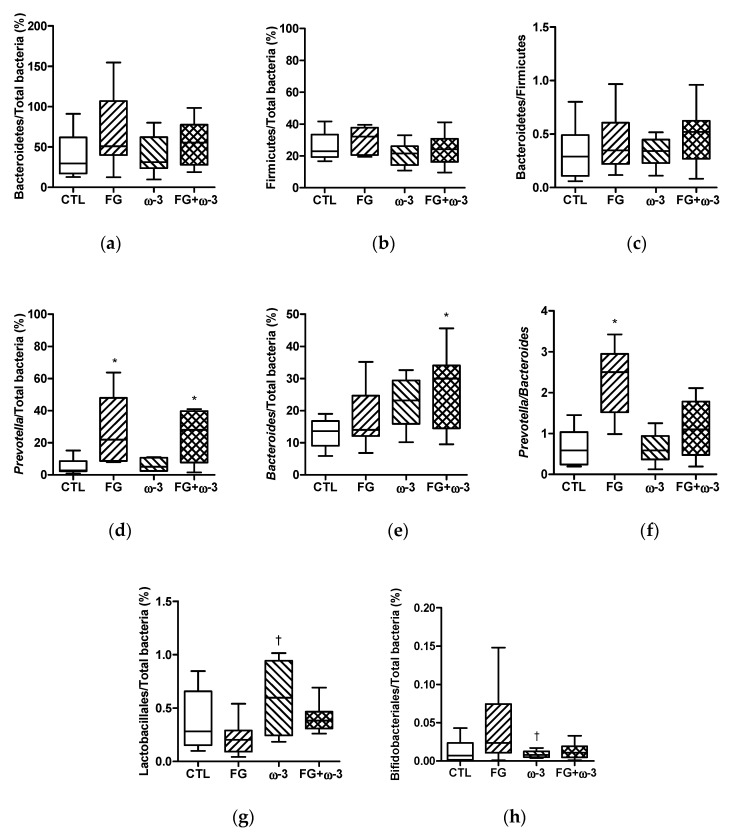
Bacteroidetes (**a**), Firmicutes (**b**), Bacteroidetes:Firmicutes ratio (**c**), Prevotella (**d**), Bacteroides (**e**), Prevotella:Bacteroides ratio (**f**), Lactobacillales (**g**) and Bifidobacteriales (**h**) in fecal samples after 21 weeks, of the different groups (CTL, FG, ω-3 and FG + ω-3) of Sprague–Dawley rats fed a standard diet. Data are presented as means with their standard error. Comparisons were made using one-way ANOVA followed by Tukey’s post-hoc test. **p* < 0.05 vs. CTL; ^†^*p* < 0.05 vs. FG.

**Table 1 nutrients-11-02606-t001:** Quantitative real-time polymerase chain reaction (PCR) primers and conditions.

Target Bacteria	Annealing Temperature (°C)	Sequences (5′-3′)	Positive Control DNA	Reference
Total Bacteria	65	F: ACT CCT ACG GGA GGC AGC AGT R: ATT ACC GCG GCT GCT GGC	(^a^)	[30]
Bacteroidetes	62	F: ACG CTA GCT ACA GGC TTA A R: ACG CTA CTT GGC TGG TTC A	*Bacteroides fragilis*	[31]
Firmicutes	52	F: CTG ATG GAG CAA CGC CGC GT R: ACA CYT AGY ACT CAT CGT TT	*Ruminococcus productus*	[32]
Lactobacillales	60	F: AGC AGT AGG GAA TCT TCC A R: CAC CGC TAC ACA TGG AG	*Lactobacillus acidophylus*	[33]
Bifidobacteriales	55	F: CTC CTG GAA ACG GGT GG R: GGT GTT CTT CCC GAT ATC TAC A	*Bifidobacterium longum*	[34]
*Bacteroides*	60	F: GGT TCT GAG AGG AGG TCC C R: GCT GCC TCC CGT AGG AGT	*Bacteroides fragilis*	[35]
*Prevotella*	60	F: CAG CAG CCG CGG TAA TA R: GGC ATC CAT CGT TTA CCG T	*Prevotella copri*	[35]

^a^ The positive control DNA used for the evaluation of Total Bacteria for each subgroup was the one selected as positive control for that subgroup.

**Table 2 nutrients-11-02606-t002:** Feed and energy intake, body weight, and plasma variables of rats supplemented with d-fagomine and/or ω-3 polyunsaturated fatty acids (PUFAs) for 23 weeks. CTL, control group; FG, group supplemented with d-fagomine; ω-3, group supplemented with ω-3 PUFAs (EPA/DHA 1:1); and FG + ω-3, group supplemented with both d-fagomine and ω-3 PUFAs.

Variables	CTL		FG		ω-3		FG + ω-3	
	Mean	SEM	Mean	SEM	Mean	SEM	Mean	SEM
Feed intake (g/day/100 g body weight)	4.6	0.5	4.8	0.4	4.6	0.5	4.9	0.4
Energy intake ^δ^ (kcal/day/100 g body weight)	13.3	1.4	13.8	1.2	13.4	1.3	14.1	1.3
Initial body weight (g)	373	7	360	3	363	7	360	7
Final body weight (g)	540	16	493*	5	523	13	497 *	11
Fasting glucose ^&^ (mg/dL)	65	2	62	1	67	2	63	2
Fasting insulin ^&^ (ng/mL)	0.56	0.10	0.34	0.03	0.65^†^	0.07	0.43	0.06
AUC from OGTT	3750	246.4	3212	246.1	3117	152.4	2674	107.5
Triglycerides (mmol/L)	0.69	0.02	0.61*	0.02	0.56***^†^	0.01	0.53***	0.02
Cholesterol (mmol/L)	3.61	0.04	3.30**	0.03	3.23***	0.08	3.24***	0.06
HDL/LDL	2.82	0.08	2.13***	0.06	2.34***	0.03	2.11***^φ^	0.04

^δ^ Energy intake is estimated as metabolizable energy based on Atwater factors: 4 kcal/g protein, 9 kcal/g fat, and 4 kcal/g available carbohydrates. ^&^ Samples from week 21. Data are presented as means with their standard errors of the mean; *n* = 9 per group. Comparisons were conducted using one-way analysis of variance (ANOVA) and Tukey’s multiple comparisons test. **p* < 0.05, ***p* < 0.01, ****p* < 0.001 vs. CTL; ^†^*p* < 0.05 vs. FG; ^φ^*p* < 0.05 vs. ω-3.

**Table 3 nutrients-11-02606-t003:** Short-chain fatty acids (SCFAs) in feces after 21 weeks. CTL, control group; FG, group supplemented with d-fagomine; ω-3, group supplemented with ω-3 PUFAs (EPA/DHA 1:1); and FG + ω-3, group supplemented with both d-fagomine and ω-3 PUFAs.

SCFAs	CTL		FG		ω-3		FG + ω-3	
	Mean	SEM	Mean	SEM	Mean	SEM	Mean	SEM
Acetic acid	115	13	125	14	23***	6	19***	4
Propionic acid	13.5	0.9	15.7	3.6	9.2	1.6	10	2
Isobutyric acid	2.7	0.2	0.9***	0.1	0.9***	0.2	0.9***	0.1
Butyric acid	17	2	27	9	12	3	16	3
Isovaleric acid	1.8	0.2	1.3	0.3	1.2	0.3	1.2	0.1
Valeric acid	1.4	0.1	1.7	0.5	1.3	0.3	1.6	0.2
Total SCFAs	152	9	158	19	48***	11	49***	7

Data are presented as means with their standard errors of the mean; *n* = 9 per group. Short-chain fatty acids (SCFAs) are given as millimoles per kilogram of feces. Comparisons were conducted using one-way ANOVA and Tukey’s multiple comparisons test. ****p* < 0.001 vs. CTL.

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
