# Peer review of "Combined Buckwheat d-Fagomine and Fish Omega-3 PUFAs Stabilize the Populations of Gut Prevotella and Bacteroides While Reducing Weight Gain in Rats"

_nutrients, 2019, doi:10.3390/nu11112606_

Round 1

Reviewer 1 Report

In the present manuscript, the authors want to explore possible complementary effects of D-Fagomine and omega-3 PUFAs on beneficial gut bacterial strains. The design experimental and methods are adequate for this kind of study. The results are interesting. My primary concern is the statistical analysis:  The design is perfect for performing a 2-way ANOVA. In this case, I suggest removing the 0-week group because it is difficult to determine if the effects are due to the time, diet (soybean oil) or other factors. Also, in the statistical analysis, I would not include the tendency since the value is higher than 0.05. This precision only adds to the confusion.

Is the soybean oil given also by oral gavage? Is it not possible to include the oil in the diet? I know that Envigo could do it. With three rats/cage, how the authors determine that each rat eats an adequate amount of food. Line 313, the authors indicate that the reduction is explained by the displacement of the pro-inflammatory omega-6 pathway towards the omega-3 pathway. Although this is possible, no data are presented to demonstrate that hypothesis.

Line 318. The combination of supplements resulted effectively in a significant reduction, but the comparison is not always with the control group (something vs the FG group).

Do you try to perform a correlation between short-chain fatty acids and the amount of the different populations? These correlations could be informative.

Author Response

In the present manuscript, the authors want to explore possible complementary effects of D-Fagomine and omega-3 PUFAs on beneficial gut bacterial strains. The design experimental and methods are adequate for this kind of study. The results are interesting.

Thanks for the positive opinion on the paper and for the constructive criticism. We hope we have now addressed all issues satisfactorily

My primary concern is the statistical analysis:  The design is perfect for performing a 2-way ANOVA. In this case, I suggest removing the 0-week group because it is difficult to determine if the effects are due to the time, diet (soybean oil) or other factors. Also, in the statistical analysis, I would not include the tendency since the value is higher than 0.05. This precision only adds to the confusion.

Following this suggestion, we have eliminated time 0 from Figure 3 and any mention to tendencies in this Figure and elsewhere, except a comment referring to perigonadal fat in section 3.1. Now we only mention the comparison with time 0 as text (section 3.4) for some selected cases with significant changes.

Is the soybean oil given also by oral gavage? Is it not possible to include the oil in the diet? I know that Envigo could do it.

We did that and the PUFAs in the feed provided by Envigo were degraded (oxidised) by 50% as evidenced by our routine quality control analysis. The company acknowledged that they could not improve their custom-made feed because they use extrusion at high temperature in the presence of regular air.

With three rats/cage, how the authors determine that each rat eats an adequate amount of food.

This is usual practise as approved by our Ethical Committee. Placing the animals in individual cages takes up to much space/resources. Most importantly, the animals are more comfortable when placed in small groups. There is always the possibility of unequal access to feed as part of the variability in such in vivo studies.

Line 313, the authors indicate that the reduction is explained by the displacement of the pro-inflammatory omega-6 pathway towards the omega-3 pathway. Although this is possible, no data are presented to demonstrate that hypothesis.

Others have provided the evidence we rely on (reference 19, review of anti-inflammatory action of omera-3 PUFAs). This information from the literature is mentioned here to explain our results with pro-inflammatory mediators.

Line 318. The combination of supplements resulted effectively in a significant reduction, but the comparison is not always with the control group (something vs the FG group). Do you try to perform a correlation between short-chain fatty acids and the amount of the different populations? These correlations could be informative.

Yes, sometimes the statistical significance is reached versus the FG group. The P values are very close to significance versus the CTL group. We present only the significant value for the sake of scientific rigour. We totally agree these correlations would be very informative if they existed. We have not found any significant correlation; therefore, there is not much we can add about it.

Reviewer 2 Report

This manuscript has been prepared with high accuracy. However, I would like to present some more detailed comments.

My general critical remark concerns the discussion of the results, in which a lot of attention was devoted to discussing the results regarding supplementation with individual supplements, i.e. D-fagomine and ῳ-3 PUFAs while Authors formulated the goal of the study as ‘to explore the possible complementary functional effects of D-fagomine and EPA/DHA 1:1 on healthy rats. Therefore, I would expect the discussion to be conducted in such a way as to show the effect of the combination of both ingredients, and then to show the differences between using their combination and a single ingredient. This would significantly strengthen the discussion, because in the Introduction the effect of individual supplements has already been confirmed.

For example: lines 310-319. The effects of the addition of EPA and DHA, also D-fagomine, were described in detail, whereas in relation to the combination of these supplements was only written “resulted in a significant reduction in the levels of all the mediators tested”. In my opinion these results require more required discussion. Were they expected? What about achieved levels of individual mediators, How these results may be explained?

Detailed comments

Line 186 – ‘were’ instead of ‘was’

Line 199 – ‘supplemented’ instead of ‘supplement’

Line 220 – ‘were statistically similar’ – I think that it should be changed e.g. ‘did not differ significantly’ or similarly

Line 219 – (data not shown) please consider including this data as supplementary data

Line 243-246 and line 271 – these sentences inform about method, therefore they should be in section Materials and Methods

Line 296 ‘in previous reports’ – references are needed.

Author Response

This manuscript has been prepared with high accuracy. However, I would like to present some more detailed comments.

 Thanks for the general opinion on the paper and the constructive comments. We hope we have now addressed all issues satisfactorily

My general critical remark concerns the discussion of the results, in which a lot of attention was devoted to discussing the results regarding supplementation with individual supplements, i.e. D-fagomine and ῳ-3 PUFAs while Authors formulated the goal of the study as ‘to explore the possible complementary functional effects of D-fagomine and EPA/DHA 1:1 on healthy rats. Therefore, I would expect the discussion to be conducted in such a way as to show the effect of the combination of both ingredients, and then to show the differences between using their combination and a single ingredient. This would significantly strengthen the discussion, because in the Introduction the effect of individual supplements has already been confirmed. For example: lines 310-319. The effects of the addition of EPA and DHA, also D-fagomine, were described in detail, whereas in relation to the combination of these supplements was only written “resulted in a significant reduction in the levels of all the mediators tested”. In my opinion these results require more required discussion. Were they expected? What about achieved levels of individual mediators, How these results may be explained?

We have now rewritten this entire section of the discussion. We have gone more deeply into the action of the supplements and we have introduced a discussion about the possible role of soybean oil added to the fagomine group. In any case, the results show that the activity of the combination most probably comes from the role of the individual components, with no synergies. This is what we can honestly say and we think it is still of interest for the readers of Nutrients

Detailed comments

Line 186 – ‘were’ instead of ‘was’

Corrected

Line 199 – ‘supplemented’ instead of ‘supplement’

Corrected

Line 220 – ‘were statistically similar’ – I think that it should be changed e.g. ‘did not differ significantly’ or similarly

Modified

Line 219 – (data not shown) please consider including this data as supplementary data

We now include the OGTT test into Table 2

Line 243-246 and line 271 – these sentences inform about method, therefore they should be in section Materials and Methods

These lines are now moved and summarized in the Materials and Methods section. Line 271 deleted

Line 296 ‘in previous reports’ – references are needed.

We have now introduced the adequate references.

Reviewer 3 Report

The current manuscript investigated Combined buckwheat D-fagomine and fish omega-3 2 PUFAs on the the populations of gut Prevotella and 3 Bacteroides, weight gain, and inflammation. This manuscript provides significant findings in this area of research and synergism effects.

The only point that I ask from authors is that data do not show any additive effect of D-fagomine and omega-3 in decreasing weight however it says in conclusion that they had complementary effects in reduction of weight gain.

Author Response

The current manuscript investigated Combined buckwheat D-fagomine and fish omega-3 2 PUFAs on the the populations of gut Prevotella and 3 Bacteroides, weight gain, and inflammation. This manuscript provides significant findings in this area of research and synergism effects.

Thanks for the positive opinion on the paper.

The only point that I ask from authors is that data do not show any additive effect of D-fagomine and omega-3 in decreasing weight however it says in conclusion that they had complementary effects in reduction of weight gain.

This is correct. We do not think that the effect on body weight is additive, it should be attributed to D-fagomine. We meant that both supplements have complementary effects “which result” in a series of outcomes that include weigh gain, not that the effects are complementary in every outcome. We have now rewritten the conclusion and hope that now the idea is more clearly presented.

Round 2

Reviewer 2 Report

Thank you for using my suggestions to improve the   manuscript . I accept them.